# High-dose oral thiamine versus placebo for chronic fatigue in patients with primary biliary cholangitis: A crossover randomized clinical trial

**Palle Bager**[1,2]*, **Lars Bossen**[1], **Rasmus Gantzel**[1], **Henning Grønbæk**[1,2]

1 Department of Hepatology and Gastroenterology, Aarhus University Hospital, Aarhus, Denmark,
2 Department of Clinical Medicine, Aarhus University, Aarhus, Denmark

* pallbage@rm.dk

## Abstract

### Background & aims

Fatigue has high negative impact on many patients with primary biliary cholangitis (PBC) and treatment options are limited. Recently we showed favorable effects of four weeks of high-dose thiamine treatment on fatigue in patients with inflammatory bowel disease. We aimed to investigate the effect and safety of high-dose (600–1800 mg daily) oral thiamine treatment on chronic fatigue in patients with PBC.

### Methods

Randomized, double-blinded, placebo-controlled crossover trial including patients with severe PBC-related fatigue. Participants were allocated 1:1 to either group 1) 4 weeks of high-dose thiamine, 4 weeks of washout, and 4 weeks of placebo; or group 2) 4 weeks of placebo, washout, and high-dose thiamine, respectively. Fatigue severity was quantified using the fatigue subscale of the PBC-40 questionnaire. The primary outcome was a fatigue reduction of $\geq$ 5 points after 4 weeks of high-dose thiamine treatment.

### Results

We enrolled 36 patients; 34 completed the study. The overall mean reduction in fatigue was 5.0 points (95% CI: 2.5 to 7.5; p < 0.001) for the combined group 1 and group 2. Crossover analysis showed a mean increase in fatigue of 0.3 points (95% CI: -4.2 to 3.8) after high-dose thiamine treatment compared to a 1.4 points (95% CI: 6.2 to –3.4) mean reduction after placebo (p = 0.55). Only mild and transient adverse events were recorded.

### Conclusion

Four weeks of high-dose oral thiamine treatment in patients with PBC was well tolerated and safe. However, high-dose thiamine was not superior to placebo in reducing PBC-related fatigue.

**Data Availability Statement:** All relevant data are within the manuscript and its Supporting Information files.

**Funding:** Helsefonden The funders had no role in study design, data collection and analysis, decision to publish, or preparation of the manuscript.

**Competing interests:** None of the authors declare competing interests regarding the present study. Henning Gronbaek received research funding from Intercept, Abbvie, NOVO Nordisk Foundation, ARLA Food for Health, and ADS AIPHIA Development Services. Consultant AstraZeneca, NOVO Nordisk, Ipsen, and data monitoring committee at CAMURUS.

## Trial registration

The trial was registered in the ClinicalTrials.gov (NCT04893993) and EudraCT (2020-004935-26).

## Introduction

Primary biliary cholangitis (PBC) is a chronic, inflammatory immune-mediated liver disease ultimately leading to liver fibrosis, cirrhosis and end-stage liver disease [1–3]. PBC is a rare disease [4] with approximately 90% being women aged between 30 to 65 years at the time of diagnosis [3]. More than 50% of patients with PBC suffer from chronic fatigue, and 20% suffer from severe fatigue with a significant negative impact on their quality of life [5–7]. However, treatment options are limited [5, 8] with liver transplantation as the only documented treatment to reduce fatigue in patients with PBC [9].

PBC-related fatigue is usually measured on a subscale of the disease-specific PBC-40 questionnaire, which includes a fatigue domain of 11 questions. All answers are scored from 1–5, providing a total fatigue score between 11 and 55 [10]. In 2013, the mean total fatigue score was 20 in the general population, and a clinical cut-off for significant PBC-related fatigue was set at a score above 32 [11].

High-dose thiamine treatment reduces fatigue in patients with inflammatory bowel disease (IBD), Parkinson's disease, and fibromyalgia [12–14]. We recently showed that high-dose oral thiamine was safe and effective in treating chronic fatigue in patients with IBD [15]. In that study, 55–75% of the patients had a significant decrease in their fatigue level and 45% of the patients reached a fatigue level similar to that of the general population. As the mechanism of action remains unknown, the favorable effects of thiamine may expand to a wider range of diseases and warrants investigations in patients with fatigue due to chronic liver disease and especially PBC where chronic fatigue is common and may be debilitating.

We hypothesized that high-dose thiamine reduces fatigue in patients with PBC and chronic fatigue. We aimed to investigate the efficacy and safety of four weeks of high-dose thiamine as a fatigue-reducing treatment in patients with PBC and severe chronic fatigue. Furthermore, we aimed to investigate the impact on quality of life measures following thiamine treatment.

## Materials and methods

### Study design

We conducted a randomized, double-blinded, placebo-controlled, crossover trial. Enrollment occurred between May 2021 and May 2022 at the Department of Hepatology and Gastroenterology at Aarhus University Hospital, Denmark. Patients were allocated 1:1 to group 1 (high-dose oral thiamine for 4 weeks in period 1, followed by 4 weeks of washout, followed by 4 weeks of oral placebo in period 2) or group 2 (oral placebo for 4 weeks in period 1, followed by 4 weeks of washout, followed by 4 weeks of high-dose oral thiamine in period 2). Individual variation in the responses was expected and minimized with the crossover design. To avoid the risk of a carryover effect, we added a 4-week washout period between period 1 and period 2. This duration was chosen as the high-dose thiamine half-life is < 5 hours and the thiamine storage time in the liver is less than 3 weeks [16, 17].

## Participants

Patients with PBC for more than 3 months, age $\geq$ 18 years, a total fatigue score on the PBC-40 questionnaire > 32 and a fatigue duration > 6 months were eligible for study inclusion [10, 11]. The cut-off point for fatigue was equivalent to the clinical cut-off point for significant fatigue in the background population (mean + 2 standard deviations) [11]. We excluded pregnant women, patients with fatigue-causing co-morbidities (i.e. uncontrolled diabetes mellitus, dysregulated thyroid disease, anemia, and vitamin D deficiency), patients with impaired kidney function (glomerular filtration rate < 60 ml/min/1.73m$^2$), patients who appeared to have low compliance, and patients who had planned surgery during the study period.

## Randomization and masking

The randomization sequence with blocks of six consisting of three randomized to group 1 and three randomized to group 2 was performed by the Hospital Pharmacy at Aarhus University Hospital. They also labelled and packed the study medicine. All investigators and participants were blinded for the treatment.

## Procedures

At the baseline visit, all included participants received tablets for the two treatment periods in two separate anonymized containers, labelled period 1 and period 2, each containing 200 tables. The thiamine tablet (SAD, Amgros I/S, Copenhagen) contained 300 mg thiamine hydrochloride. Placebo tablets were produced and delivered by Herlev Hospital Pharmacy, The Capital Region of Denmark. The daily dose depended on body weight (BW) and gender (female/male) according to the scheme: BW < 60 kg: 600/900 mg (2/3 tablets), BW 60–70 kg: 900/1200 mg (3/4 tablets), BW 71–80 kg: 1200/1500 mg (4/5 tablets), and BW > 80 kg: 1500/1800 mg (5/6 tablets). The dosing scheme was inspired by our recent trial in patients with IBD [15]. At the week 12 (end of trial) visit, the investigator counted the remaining tablets. Adherence was defined as >80% of the schemed study medication taken. The participants were reminded not to ingest any additional thiamine during the trial period and advised not to take the study drug in the evening to prevent temporary sleeplessness. At the week 12 visit the participants were asked to guess in which period they received thiamin tablets and placebo respectively.

Study visits in the clinic were scheduled at baseline and week 12 (end of trial), with the completion of questionnaires, drawing of blood samples, and safety assessments. At week 4 and week 8, the patients were given instructions electronically and completed the questionnaires directly in a REDCap database [18].

Fatigue was estimated using the fatigue sub-scale on the PBC-40 questionnaire [10] covering a domain of 11 questions where responses are given on a 5-point Likert scale, ranging from 1 'not at all' to 5 'very much'. Consequently, the total fatigue score range is 11–55. The PBC-40 questionnaire is validated and translated into Danish.

The Danish version of the EQ-5D-5L was used to measure generic health-related quality of life (HRQoL). The EQ-5D-5L has been thoroughly validated (see also www.euroqol.org). The EQ-5D-5L classification system comprises five dimensions (mobility, self-care, usual activities, pain/discomfort, and anxiety/depression), each with five levels. In addition, the EQ-5D-5L instrument yields a single overall health index score based on a visual analog scale (VAS) from 0–100 with high scores indicating high self-reported overall HRQoL [19].

Adverse events were registered and reported to the Ethics Committee in Central Denmark Region and the Danish Medical Agency.

## Outcomes

The primary endpoint was a reduction of $\geq 5$ points in the total fatigue score after 4 weeks of thiamine treatment compared to placebo. A reduction of $\geq 5$ points was defined as clinically relevant since this would correspond to a ~10–15% reduction in the fatigue score among all included patients with possible fatigue scores between 33 and 55. The secondary endpoints covered the change in fatigue score at week twelve and changes in HRQoL at week 4, 8 and 12.

## Statistical analysis

This 2 x 2 crossover study required 15 patients in each group to detect a mean difference between thiamine and placebo. This was based on the following assumptions: (i) a decrease of $\geq 5$ points on the PBC-40 fatigue sub-scale, (ii) a standard deviation of 6.6 points based on published data for fatigue in the background population (11), (iii) an alpha value of 0.05, and a power of 80% [20]. This program was used for calculation: http://hedwig.mgh.harvard.edu/sample_size/size.html#cross.

We estimated approximately 10% drop-out demanding a total sample size of 36 patients.

Crossover analysis for fatigue scores averaged the between-treatment difference for each patient within each treatment period and then across both treatment periods, providing an estimate of treatment effect [21, 22]. The estimated treatment difference, 95% confidence interval (CI) and p-values were adjusted for period and sequence effects in the variance model analysis. To check for a potential carryover effect, we analyzed the within-subject sums of the results from both periods and compared the two groups, using an unpaired t-test. Subsequently, we did a parallel group comparison considering group 2 as subject to delayed thiamine treatment (period 2). In our primary analytical approach per the crossover analysis, group 1 was considered to be in the thiamine treatment group at week 4 and in the control group at week 12, whereas group 2 was considered to be in the control group at week 4 and in the thiamine treatment group at week 12. Differences in fatigue between week 4 and week 12 were compared between groups, using an unpaired t-test. Sensitivity analysis excluding non-adherent participants was performed. We aimed to do both intention-to-treat (ITT) analysis and per-protocol analysis.

Study data were managed using the REDCap data capture tool hosted at Aarhus University [18]. Data analysis was conducted in Stata.

The trial was approved by the Ethics Committee in Central Denmark Region (j.no. 76946) and the Danish Medical Agency (EudraCT j.no. 2020-004935-26). The trial was conducted in accordance with the Declaration of Helsinki and the GCP principles and was monitored by the GCP unit at Aarhus and Aalborg University Hospitals. The trial was registered in the ClinicalTrials.gov (NCT04893993).

## Results

Thirty-six of 74 patients with PBC and chronic fatigue were assessed for eligibility, enrolled and randomly assigned to the treatment sequences (Fig 1). In brief, the fourteen patients who were excluded due to fatigue-causing comorbidity included: fibromyalgia, dysregulated diabetes, cancer, apoplexia, anemia, Sjögren's Syndrome, vitamin D deficiency, and combinations hereof.

The baseline characteristics are shown in Table 1. The median fatigue score was 39 in group 1 and 40 in group 2.

Two participants dropped out during period 1; one from group 1 due to SARS-CoV-2 infection and one from group 2 who withdrew informed consent. None of the drop-outs were related to study medication. In the following analysis, we used data from the 34 patients who

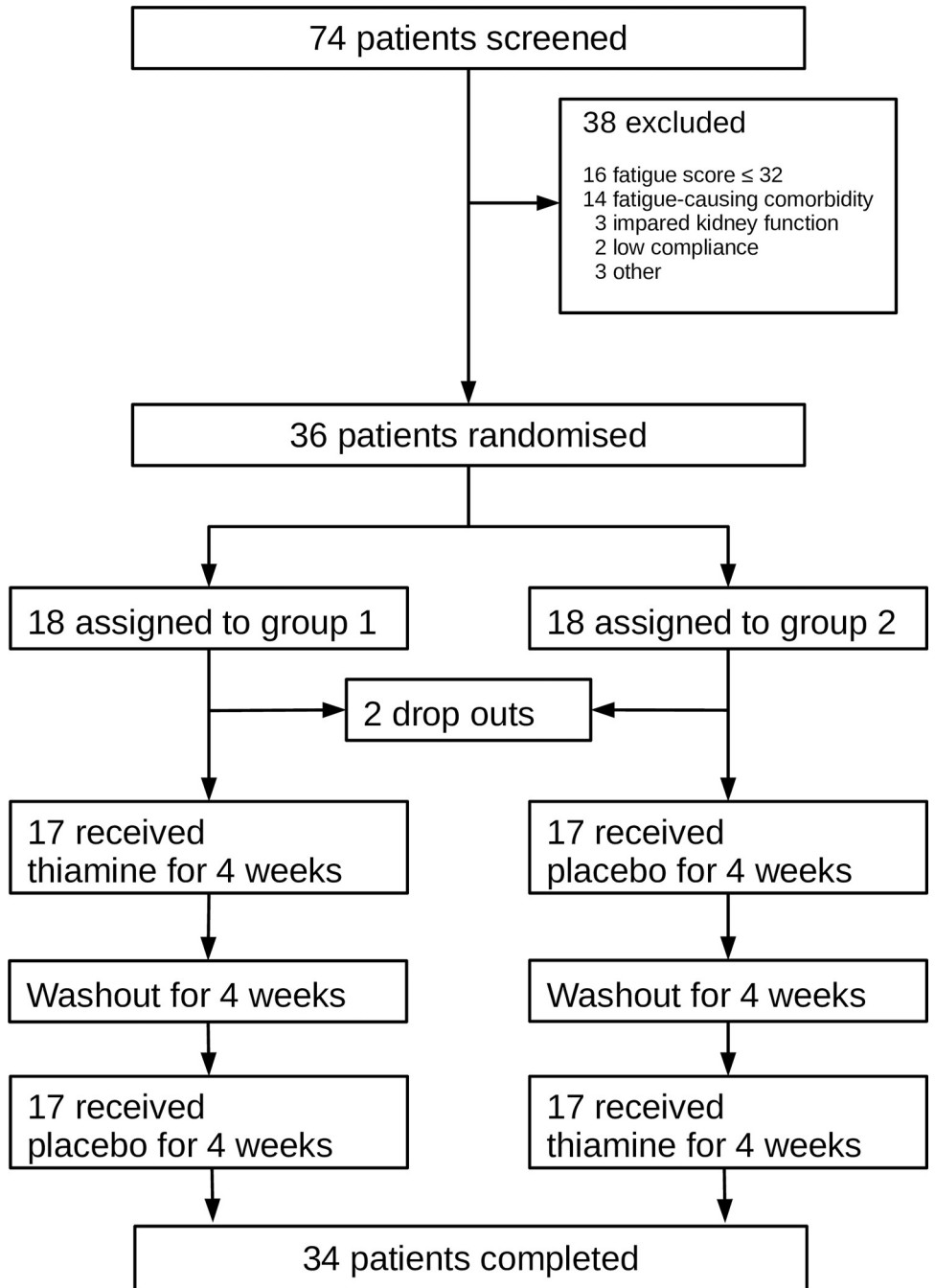

**Fig 1. Consolidated standards of reporting trials diagram.**

completed the study. Due to missing data from the drop-outs, we performed per-protocol analysis and abstained from doing ITT analysis.

The overall mean reduction of fatigue in the PBC-40 fatigue subscale score was 5.0 points (95% CI 2.5 to 7.5; p < 0.001) for the combined group 1 and group 2. Both groups increased by 2 points in fatigue score during the washout period. Given this and a confirmatory analysis showing a non-significant inter-group difference in the sum of fatigue at week 4 and week 12,

**Table 1. Baseline characteristics.**

| | Group 1: | Group 2: |
|---|---|---|
| | thiamine followed by placebo (n = 18) | placebo followed by thiamine (n = 18) |
| Demographics | | |
| Age, years, mean (SD) | 56 (10) | 61 (9) |
| Sex, female, n (%) | 16 (89) | 17 (94) |
| Body Mass Index, mean (SD) | 29 (7) | 27 (6) |
| Disease, n (%) | | |
| AIH overlap | 2 (12) | 2 (12) |
| Liver cirrhosis | 1 (6) | 1 (6) |
| Biochemistry, median (IQR) | | |
| ALAT (U/l) | 40 (29–65) | 32 (27–45) |
| Bilirubin (μmol/l) | 11 (8–13) | 8 (6–10) |
| Alkaline phosphatase (U/l) | 134 (94–155) | 128 (101–175) |
| FIB-4 score (U/l) | 1.2 (1.0–1.5) | 1.2 (0.9–1.8) |
| CRP (mg/l) | 4 (1–8) | 4 (1–9) |
| sCD163 (mg/l) | 3.2 (2.8–4.0) | 3.4 (3.0–3.7) |
| WBC ($10^9$/l) | 6 (2) | 7 (2) |
| Haemoglobin (g/dl) | 13.7 (12.7–14.0) | 14.0 (13.4–15.4) |
| Platelet count ($10^9$/l) | 253 (218–293) | 273 (235–284) |
| Comorbidity, n (%) | | |
| Osteoporosis | 4 (24) | 6 (35) |
| Diabetes | 2 (12) | 2 (12) |
| Hypertension | 3 (18) | 2 (12) |
| Medication, n (%) | | |
| UDCA | 18 (100) | 18 (100) |
| Bezafibrate | 6 (35) | 0 (0) |
| Obethicolic acid | 0 (0) | 0 (0) |
| Fatigue and HRQoL, median (IQR) | | |
| PBC-40, fatigue sub-scale | 39 (37–42) | 40 (37–41) |
| EQ-5D-5L VAS | 65 (50–75) | 75 (66–80) |

Abbreviations: AIH, autoimmune hepatitis; CRP, C-reactive protein; WBC, white blood cells; UDCA, ursodeoxycholic acid; EQ.5D-5L, European Quality of life scale; SD, standard deviation; IQR, interquartile range.

we assumed no carryover effect. Therefore, we assessed the delayed treatment model. The crossover analysis showed a mean increase in fatigue of 0.3 points (95% CI: -4.2 to 3.8) after high-dose thiamine treatment compared to a 1.4 points (95% CI: 6.2 to –3.4) mean reduction after placebo (p = 0.55).

Furthermore, no statistical difference in HRQoL measures were detected in the crossover analysis (p = 0.90) (Table 2).

Forty-seven percent (n = 8) in group 1 and 29% (n = 5) in group 2 experienced an improvement ≥ 5 points while on thiamine treatment compared with 41% (n = 7) and 24% (n = 4) while on placebo, respectively (p = 0.71).

Fig 2 illustrates the course of fatigue scores during the study period for both groups.

At baseline six patients received Bezafibrate in Group 1 compared to none in Group 2. Supplementary analysis stratifying for Bezafibrate did not substantially change the main results. One patient in each group was < 80% adherent to the treatment when receiving thiamine.

**Table 2. Crossover analysis of fatigue and health-related quality of life.**

| | | Group 1: thiamine followed by placebo (n = 17) | | | Group 2: placebo followed by thiamine (n = 17) | | | |
|---|---|---|---|---|---|---|---|---|
| | | Week 4 | Week 12 | Difference | Week 4 | Week 12 | Difference | p–value |
| Fatigue | | | | | | | | |
| | PBC40 • fatigue subscale | 34.6 (7.1) | 33.2 (7.5) | -1.4 (2.3) | 35.5 (7.2) | 35.8 (5.3) | 0.3 (1.9) | 0.55 |
| HRQoL | | | | | | | | |
| | EQ5D VAS • range 0–100 | 68.3 (13.0) | 69.5 (18.7) | 1.2 (5.0) | 69.1 (15.4) | 70.2 (13.2) | 1.1 (4.1) | 0.90 |

*Note*: Data are mean, (SD).

Abbreviations: EQ5D VAS, European Quality of life, VAS.

Sensitivity analysis excluding the two non-adherent participants showed an overall mean reduction of fatigue in the PBC-40 fatigue subscale score of 5.4 points (95% CI 2.8 to 7.2; $p < 0.001$) for the combined group 1 and group 2. In addition, crossover analysis showed a mean reduction in fatigue of 0.1 points (95% CI: -4.2 to 4.4) after high-dose thiamine treatment compared to a 1.8 points (95% CI: -3.8 to 6.8) mean reduction after placebo (p = 0.60).

Additional analysis found no correlations between the participants' characteristics and reduction in fatigue in the study period. This also includes the individual baseline cognitive symptoms sub-score on the PBC-40. Upon unblinding, it was revealed that 20 participants (58%) correctly surmised the treatment allocation.

Adverse events (AEs) and serious adverse events (SAEs) were monitored continuously during the study. No SAEs related to the study drug were detected. Three SAEs occurred and required brief hospital admissions, including SARS-CoV-2 infection (placebo-period), other infection (thiamine-period), and exacerbation of chronic obstructive pulmonary disease (washout-period). None of the SAEs were judged as related to the study drug. A few temporary AEs were recorded including sore throat/cold symptoms (3 in the thiamine-period and 2 in the placebo-period), elevated ALAT (1 in the placebo-period), fall on a pavement (1 in the placebo-period), and angina pectoris event (1 in the thiamine-period).

## Discussion

There is an unmet need for new and better treatments of fatigue in PBC patients. To our knowledge, this is the first trial to investigate the safety and efficacy of high-dose oral thiamine on chronic fatigue in patients with PBC. In this study four weeks of high-dose oral thiamine was well tolerated and safe in patients with PBC. However, we could not demonstrate any significant effect of high-dose thiamine on chronic fatigue compared to placebo. These results diverge from the recent study in patients with IBD where four weeks of high-dose oral thiamine was superior to placebo in reducing chronic fatigue [15].

The mechanisms behind the effect of thiamine on fatigue are still unsettled [23–25]. Thus, etiology of chronic fatigue in patients with PBC may differ from that of chronic fatigue in patients with IBD. We observed no differences in the effect when stratifying for self-reported cognitive symptoms, as suggested to be a factor for fatigue in patients with PBC [26]. Neither did we reveal any predictors for effect from high-dose thiamine.

Fatigue remains a burdensome companion for many patients with PBC and interventions that can reduce fatigue are highly needed [27]. This could be of any origin. Small scale studies

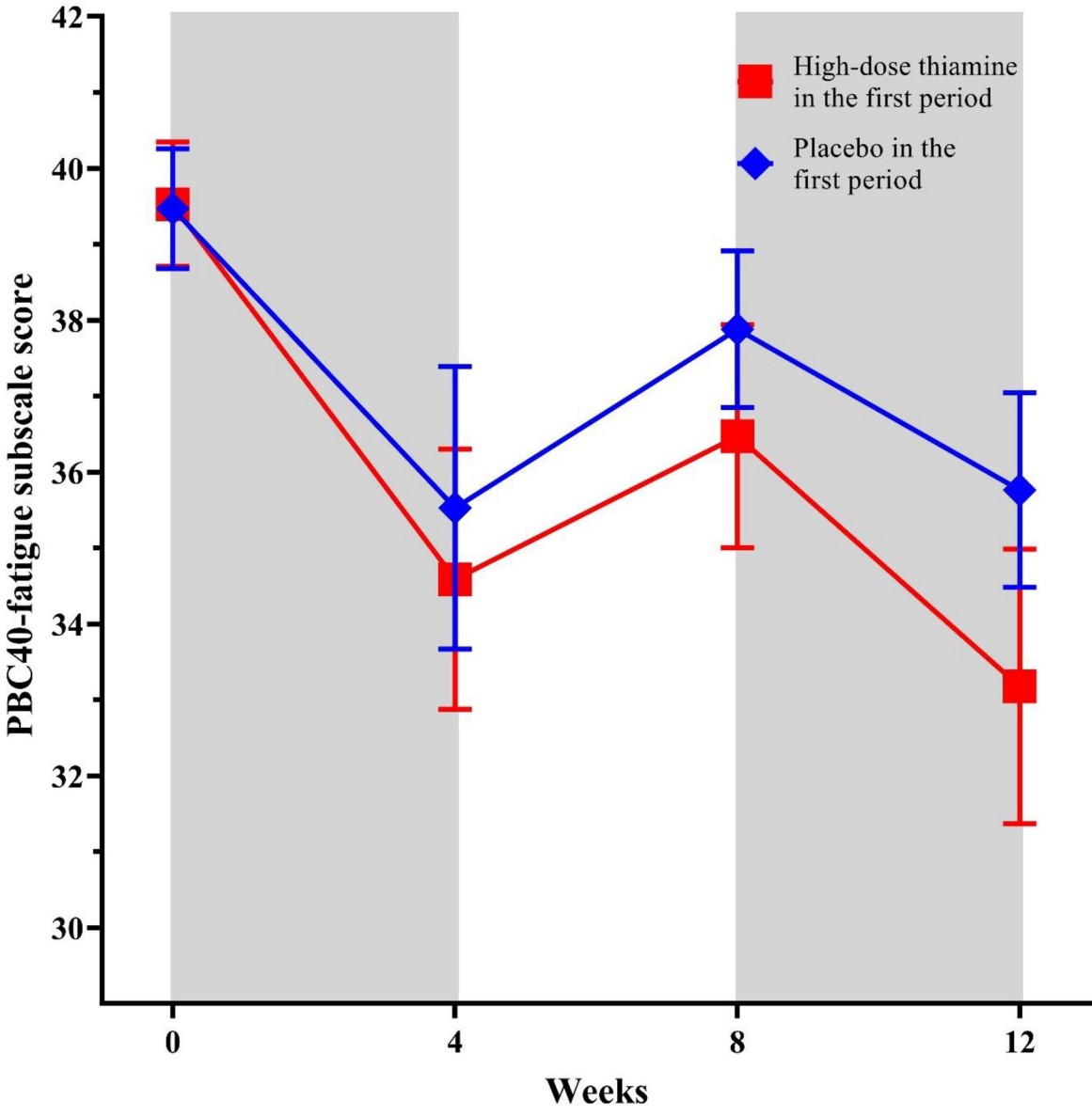

**Fig 2. Fatigue scores between groups.**

on pharmaceutical products like seladelpar [28] and S-adenosyl-L-methionine [29] have been performed and showed effect on HRQoL, including fatigue. However, larger studies are needed [27]. According to the clinicaltrials.gov database, two non-pharmacological interventions have been identified. A mindfulness study planned in the US is terminated, while a home exercise study has been completed in the UK [30] but not yet reported. In addition to intervention studies, there is a need for research that can contribute to a better understanding of the pathogenesis and mechanisms of fatigue in PBC (and more broadly). Given that fatigue is a non-specific symptom observed in a broad range of diseases, exploring fatigue-alleviating solutions from other diseases in a PBC population could be valuable [31]. Specifically, interventions tested in other autoimmune diseases are worth investigating. The present study demonstrates how findings from IBD were applied within a PBC population.

Despite the strong design of this study, it has some limitations. Firstly, even though we did a straight forward power calculation a priory accounting for drop-outs the study might be underpowered. A larger study may show effect of high-dose thiamine. However, a pragmatic implication of our finding could be to consider refraining from any further studies on PBC-related fatigue and high-dose thiamine. Furthermore, it is known that the smell and taste of vitamins can be intense, potentially posing a challenge to the blinding of the study. However, only 58% of the participants surmised the correct treatment allocation, which indicates a reliable blinding of the study drug.

In conclusion, this clinical study showed that four weeks of high-dose oral thiamine treatment was similar to placebo in reducing fatigue in patients with PBC and chronic fatigue. The thiamine treatment was safe and well tolerated by the patients. Further studies are needed to help the group of patients with PBC suffering from chronic fatigue.

## Supporting information

**S1 Checklist. CONSORT 2010 checklist of information to include when reporting a randomised trial\*.**
(PDF)

**S1 Data.**
(XLSX)

**S1 File.**
(PDF)

## Acknowledgments

We thank the GCP unit at Aarhus and Aalborg University Hospital, Denmark, for advice and rigorous monitoring. The study was partly funded by donations from Helsefonden.

## Author Contributions

**Conceptualization:** Palle Bager, Lars Bossen, Rasmus Gantzel, Henning Grønbæk.

**Data curation:** Palle Bager, Lars Bossen, Rasmus Gantzel.

**Formal analysis:** Palle Bager, Lars Bossen, Rasmus Gantzel.

**Funding acquisition:** Lars Bossen, Henning Grønbæk.

**Investigation:** Palle Bager, Lars Bossen, Rasmus Gantzel, Henning Grønbæk.

**Methodology:** Palle Bager, Lars Bossen, Rasmus Gantzel, Henning Grønbæk.

**Project administration:** Palle Bager, Lars Bossen, Henning Grønbæk.

**Resources:** Palle Bager, Lars Bossen, Rasmus Gantzel, Henning Grønbæk.

**Software:** Palle Bager, Rasmus Gantzel.

**Supervision:** Henning Grønbæk.

**Visualization:** Palle Bager, Rasmus Gantzel.

**Writing – original draft:** Palle Bager, Lars Bossen, Rasmus Gantzel.

**Writing – review & editing:** Palle Bager, Lars Bossen, Henning Grønbæk.

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
