## [Decision Letter · Decision Letter 0]

2 Jan 2024

PONE-D-23-41263Please find enclosed a manuscript for PLOS ONE entitled: 

High-dose oral thiamine versus placebo for chronic fatigue in patients with primary biliary cholangitis: A crossover randomized clinical trialPLOS ONE

Dear Dr. Bager,

Thank you for submitting your manuscript to PLOS ONE. After careful consideration, we feel that it has merit but does not fully meet PLOS ONE’s publication criteria as it currently stands. Therefore, we invite you to submit a revised version of the manuscript that addresses the points raised during the review process.

We look forward to receiving your revised manuscript.

Kind regards,

Wan-Long Chuang, M.D., Ph.D.

Academic Editor

PLOS ONE

Journal Requirements:

 https://doi.org/10.1111/apt.16166

In your revision ensure you cite all your sources (including your own works), and quote or rephrase any duplicated text outside the methods section. Further consideration is dependent on these concerns being addressed.

   "Helsefonden"  

5. In the online submission form, you indicated that "Data can be shared upon reasonably request". 

Reviewers' comments:

Reviewer's Responses to Questions

**Comments to the Author**

1. Is the manuscript technically sound, and do the data support the conclusions?

Reviewer #1: Yes

Reviewer #2: Yes

Reviewer #3: Partly

2. Has the statistical analysis been performed appropriately and rigorously? 

Reviewer #1: Yes

Reviewer #2: Yes

Reviewer #3: Yes

3. Have the authors made all data underlying the findings in their manuscript fully available?

Reviewer #1: Yes

Reviewer #2: No

Reviewer #3: No

4. Is the manuscript presented in an intelligible fashion and written in standard English?

Reviewer #1: Yes

Reviewer #2: No

Reviewer #3: Yes

5. Review Comments to the Author

Reviewer #1: This randomized, double-blinded, and placebo-control study investigated the effect of thiamin in treating fatigue in PBC patients. Four-week thiamin treatment was administered before or after placebo. PBC-40 was used to assess the changes of fatigue. The study concluded that thiamin administration did not improve fatigue when compared with placebo. Overall, this study was well-written and scientifically sound.

Comments

1. The limitation of study is the patient number. Owing to lack of reference study, larger number of patients would be helpful to validate the treatment efficacy.

2. In group 1, more patients took Bezafibrate that had beneficial effect in fatigue symptom. It would compromise the analysis, although there was negative result of thiamin treatment.

3. Many conditions may also contribute to fatigue. No further description was noted.

4. How to explain the improvement of PBC-40 score in placebo and similar patterns of score changes during entire study period of both groups? true double-blinded?

Reviewer #2: General comments:

This manuscript presents the results of a crossover trial of oral thiamine treatment on chronic fatigue in participants with primary biliary cholangitis. The manuscript is well-written and easy to follow. As far as I can tell, all necessary sections and components are in place.

Given this is a small trial, the small sample size limits the statistical methods choices. Thus, I found this manuscript straightforward, and don't feel I have much to offer to improve this manuscript. In my specific comments below I have a few suggestions on changes that could be made.

Specific comments:

1. The most limiting factor, at least when trying to assess efficacy, is the sample size. I found the sample size acceptable but on the low end for a phase 1 safety trial, but very much lacking to estimate efficacy with any robustness. Thus, I'm not sure how to feel about the efficacy results. Is the aim of the efficacy results proof of concept? I fully understand the safety results based on the justification in the introduction, but the efficacy results feel as though they were doomed from the start and I'm not sure whether the efficacy analyses meet criterion #3 (https://journals.plos.org/plosone/s/criteria-for-publication).

2. (lines 148-152) I suggest including a citation or some information on the formula or program that was used to calculate the sample size estimate.

3. (line 149) You may also want to explain a little bit about the choice of at least of 5 point decrease on the fatigue subscale. I operationalized that as dropping a point on 5 of the 11 questions. That seems reasonable to me, though it's not a scale I know well.

4. (Table 1) I suggest including standardized mean differences (but not p-values) to assess the differences between the groups.

Reviewer #3: Bager et al. conducted a crossover randomized clinical trial to investigate the effect and safety of high dose oral thiamine treatment on chronic fatigue in patients with PBC. They found that high dose thiamine treatment was safe but was not superior to placebo in reducing PBC-related fatigue.

Major comments

1. The primary outcome described in the abstract and the methods section are not exactly the same. Please clarify.

2. What is the primary analysis of this study (intention-to-treat analysis or per-protocol analysis or both)? Please clarify. It is suggested to present the results of both intention-to-treat and per-protocol analysis in the main text or in the supplementary file.

3. How to determine the case numbers in each group before the trial?

4. How to diagnose the primary biliary cholangitis? What’s the criteria? The proportion of liver cirrhosis in these study is very low, what’s the reason? Does the low ratio of cirrhosis implicate selection bias?

5. What’s the dose of URSO and what kind of patients received the bezafibrate? The reasons of fourteen patients with fatigue-causing comorbidity should be disclosed.

6. In the results section, the authors stated that sensitivity analysis excluding the two non-adherent participants did not change the results. However, there is no relevant statement describing the process of sensitivity test in the methods section. Please add it. In addition, please also present the data of sensitivity test in the main text or in the supplementary file.

7. In the results section, the authors mentioned three SAEs occurred and a few temporary AEs were recorded. Please list in a table how many patients in each group experienced each AE and SAE.

8. In Figure 1, two patients were excluded in the screening process due to low compliance. However, “low compliance” is not an exclusion criterion in the methods section. Please clarify.

9. In Figure 1, three patients were excluded in the screening process due to other reasons. Please elaborate on the reasons.

Minor comments

1. Table 2: in the column of Group 1, the difference of EQ5D VAS between Week 12 and Week 4 is presented as “2.2”. Should the data be corrected to “1.2” (69.5-68.3=1.2)? Please clarify.

2. Figure 1: a spelling error is noted. Should “impaired kidney funktion” be corrected to “impaired kidney function”? Please clarify.

6. PLOS authors have the option to publish the peer review history of their article (what does this mean?). If published, this will include your full peer review and any attached files.

Reviewer #1: No

Reviewer #2: No

Reviewer #3: No

---

## [Author Response · Author response to Decision Letter 0]

31 Jan 2024

Authors response to the reviewers

Reviewer #1: 

Comment 1:

The limitation of study is the patient number. Owing to lack of reference study, larger number of patients would be helpful to validate the treatment efficacy. 

Response: 

We partly agree that number of patients included may be a limitation. However, the number of patients were estimated based on a specific power calculation based on our data from IBD patients. A larger study population may have revealed some effect. However, our results were not close to show a statistically significant difference. Based on the obtained results a power calculation for larger study would have demanded approximately 140 patients. Assumptions: 1) a decrease of ≥ 5 points on the PBC-40 fatigue sub-scale, 2) a within patient standard deviation of 12 points, 3) an alpha value of 0.05, and a power of 90% 4) approximately 10-15 % drop-outs. However, as we write in the discussion, the study might have been underpowered; but we also suggest that thiamine as a treatment option for PBC-fatigue could be rejected as a result of our findings.

Comment 2: 

In group 1, more patients took Bezafibrate that had beneficial effect in fatigue symptom. It would compromise the analysis, although there was negative result of thiamin treatment.

Response:

Thank you for this observation. However, supplementary analysis stratifying for Bezafribrate showed similar results. We have added a sentence regarding this in the Results section (page 12, lines 205-7). 

Comment 3:

Many conditions may also contribute to fatigue. No further description was noted.

Response: 

We agree and since fatigue can be multifactorial, we excluded patients who had obvious other reasons for fatigue than PBC and the most frequent reasons are now mentioned in the method section. In addition, the RCT design aim to equally distribute participants who might have other reasons for fatigue.

Comment 4:

How to explain the improvement of PBC-40 score in placebo and similar patterns of score changes during entire study period of both groups? true double-blinded?

Response: 

Thank you for this relevant comment. The most obvious explanation is that the effect of thiamine was not superior to placebo. It is no surprise that both groups had a decrease in fatigue-scores in the first period, as the participants was hoping for an effect. In the first period, none of the participants had tried any of the blinded treatments before. Based on individual experiences in period one, some participants would probably be able to guess the random allocation when exposed for the study drug in period two. This could be due to taste, smell of urine or due to the effect on fatigue. However, as mentioned in the discussion, only 58% guessed their allocation sequence correctly indicating a successful blinding.

Reviewer #2: 

Comment 1: 

The most limiting factor, at least when trying to assess efficacy, is the sample size. I found the sample size acceptable but on the low end for a phase 1 safety trial, but very much lacking to estimate efficacy with any robustness. Thus, I'm not sure how to feel about the efficacy results. Is the aim of the efficacy results proof of concept? I fully understand the safety results based on the justification in the introduction, but the efficacy results feel as though they were doomed from the start and I'm not sure whether the efficacy analyses meet criterion #3 (https://journals.plos.org/plosone/s/criteria-for-publication).

Response: 

Please see our response to reviewer 1 (Q1) above. 

In addition, we agree that we may have been too optimistic regarding the ‘expected’ efficacy of thiamine vs placebo (i.e. we expected ≥5 points improvement from thiamine compared to placebo). On the other hand, a 5-point reduction is between 10-15% of the baseline fatigue score and less would – in our opinion – not be clinically relevant. Further, our optimism was based on the convincing results from the Inflammatory Bowel Disease-Fatigue-Thiamine study. Moreover, as stated in Q1, our results did not suggest a difference between thiamine and placebo, hence the missing efficacy was most likely just because thiamine does not improve fatigue in PBC patients. 

Comment 2:

(lines 148-152) I suggest including a citation or some information on the formula or program that was used to calculate the sample size estimate. 

Response:

We used this web-service for calculations: http://hedwig.mgh.harvard.edu/sample_size/size.html#cross

This has also been added to the manuscript (page 9, line 155).

Comment 3:

(line 149) You may also want to explain a little bit about the choice of at least of 5 point decrease on the fatigue subscale. I operationalized that as dropping a point on 5 of the 11 questions. That seems reasonable to me, though it's not a scale I know well. 

Response:

Thanks for pointing this out. We have added a sentence about this under ‘outcomes’ in the methods section and provide a deeper explanation here: We do not know of any similar previous studies, and thus we had to choose our own clinically relevant outcome. We chose a 5-point reduction for two reasons: 1) The possible scores on the fatigue subscale ranges between 33-55, as the minimum score for significant fatigue is 33. Thus a 5-point reduction equals a ~10-15% reduction in the fatigue score which we argue is clinically relevant; 2) Based on data collected in other clinical PBC studies at our site (data not published), we expected the median fatigue score to be around 38 among the included patients. Thus, a 5-point reduction would bring almost 50% of the patients into the spectrum of ‘normal fatigue’. 

Comment 4: 

(Table 1) I suggest including standardized mean differences (but not p-values) to assess the differences between the groups. 

Response: 

Most data in Table 1 are reported as median (IQR). The reason for this is that some of the biochemistry data doesn't have a normal distribution. We can change the presentation of data in Table and add differences (+/- p-values). However, we believe that the current presentation of data will give the best overview. 

Reviewer #3: 

Comment 1:

The primary outcome described in the abstract and the methods section are not exactly the same. Please clarify.

Response: 

Good observation. The wording in 'outcomes' in the 'M+M' section has been changed. 

Comment 2: 

What is the primary analysis of this study (intention-to-treat analysis or per-protocol analysis or both)? Please clarify. It is suggested to present the results of both intention-to-treat and per-protocol analysis in the main text or in the supplementary file.

Response: 

Good point. Due to missing data, we only performed per-protocol analysis. We have added text to both the 'Statistics' and the 'Results' sections.

Comment 3: 

How to determine the case numbers in each group before the trial?

Response: 

Please see our response to reviewer 2 (Q1) above.

Comment 4: 

How to diagnose the primary biliary cholangitis? What’s the criteria? The proportion of liver cirrhosis in this study is very low, what’s the reason? Does the low ratio of cirrhosis implicate selection bias? 

Response:

Thank you for this relevant comment. Primary biliary cholangitis was diagnosed in accordance with international consensus diagnostic criteria (reference #5 'EASL Clinical Practice Guidelines: The diagnosis and management of patients with primary biliary cholangitis. J Hepatol. 2017'). In the present study, 2 patients (6%) had cirrhosis, which is slightly lower than in a recent Danish study reporting a cirrhosis prevalence of 13% among patients with PBC. This may be due to our exclusion criteria including impaired kidney function and anaemia – two common complications among patients with cirrhosis. 

Comment 5: 

What’s the dose of URSO and what kind of patients received the bezafibrate? The reasons of fourteen patients with fatigue-causing comorbidity should be disclosed. 

Response: 

A) In Denmark, standard dose of UDCA was 750mg/day up until the introduction of bezafibrate (around 2019 in Denmark, and thus before the start of inclusion into this study). Since then, we have continued on 750mg/day if patients were sufficiently treated, i.e. their ALP was below 1.5XULN (160 I/U). If not, they were increased to 10-15mg/kg/day, and if that did not bring ALP below 160 I/U we have initiated treatment with bezafibrate. Another indication for bezafibrate treatment was progression on either liver stiffness (transient elastography) or liver fibrosis (biopsy). The last indication is based on an individual clinical judgement from hepatologists.

B) In brief, the fourteen patients who were excluded due to fatigue-causing comorbidity included: fibromyalgia, dysregulated diabetes, cancer, apoplexy, anemia, Sjögren's Syndrome, vitamin D deficiency, and combinations hereof. This sentence has also been added to the manuscript (page 10, line 180-2).

Comment 6: 

In the results section, the authors stated that sensitivity analysis excluding the two non-adherent participants did not change the results. However, there is no relevant statement describing the process of sensitivity test in the methods section. Please add it. In addition, please also present the data of sensitivity test in the main text or in the supplementary file.

Response: 

The analysis excluding the 'non-adherent' patients have now been described both in the method- and the results section. 

Comment 7: 

In the results section, the authors mentioned three SAEs occurred and a few temporary AEs were recorded. Please list in a table how many patients in each group experienced each AE and SAE. 

Response:

Events are now followed by description of treatment-group in brackets in the text. 

Comment 8: 

In Figure 1, two patients were excluded in the screening process due to low compliance. However, “low compliance” is not an exclusion criterion in the methods section. Please clarify.

Response: 

Good point. This exclusion criterion has been added to the manuscript text. 

9. In Figure 1, three patients were excluded in the screening process due to other reasons. Please elaborate on the reasons.

Response:

Sure. Two cases had fatigue for other reasons (than comorbidity) and one was newly diagnosed with PBC. 

Minor comment 1:

Table 2: in the column of Group 1, the difference of EQ5D VAS between Week 12 and Week 4 is presented as “2.2”. Should the data be corrected to “1.2” (69.5-68.3=1.2)? Please clarify.

Response: 

Thank you for identifying this typing error. The correct number is 1.2 and has been changed in Table 2.R1 

Minor comment 2: 

Figure 1: a spelling error is noted. Should “impaired kidney funktion” be corrected to “impaired kidney function”? Please clarify. 

Response: 

Thank you. The spelling has been corrected.

---

## [Decision Letter · Decision Letter 1]

18 Feb 2024

PONE-D-23-41263R1High-dose oral thiamine versus placebo for chronic fatigue in patients with primary biliary cholangitis: A crossover randomized clinical trialPLOS ONE

Dear Dr. Bager,

Thank you for submitting your manuscript to PLOS ONE. After careful consideration, we feel that it has merit but does not fully meet PLOS ONE’s publication criteria as it currently stands. Therefore, we invite you to submit a revised version of the manuscript that addresses the points raised during the review process.

We look forward to receiving your revised manuscript.

Kind regards,

Wan-Long Chuang, M.D., Ph.D.

Academic Editor

PLOS ONE

Reviewers' comments:

Reviewer's Responses to Questions

**Comments to the Author**

1. If the authors have adequately addressed your comments raised in a previous round of review and you feel that this manuscript is now acceptable for publication, you may indicate that here to bypass the “Comments to the Author” section, enter your conflict of interest statement in the “Confidential to Editor” section, and submit your "Accept" recommendation.

Reviewer #1: (No Response)

Reviewer #2: All comments have been addressed

Reviewer #3: All comments have been addressed

2. Is the manuscript technically sound, and do the data support the conclusions?

Reviewer #1: No

Reviewer #2: (No Response)

Reviewer #3: Yes

3. Has the statistical analysis been performed appropriately and rigorously? 

Reviewer #1: Yes

Reviewer #2: (No Response)

Reviewer #3: Yes

4. Have the authors made all data underlying the findings in their manuscript fully available?

Reviewer #1: Yes

Reviewer #2: (No Response)

Reviewer #3: Yes

5. Is the manuscript presented in an intelligible fashion and written in standard English?

Reviewer #1: Yes

Reviewer #2: (No Response)

Reviewer #3: Yes

6. Review Comments to the Author

Reviewer #1: This manuscript may report as a preliminary results of clinical trial rather than a complete study. It needs more patients enrolled for consolidate findings.

Reviewer #2: (No Response)

Reviewer #3: Although the authors had replied the questions raised by the reviewers, there are still some points that needed to be clarified or revised.

1. The intervention of this study is supplementation of high dose thiamine for four weeks. The authors have explained the half life and storage time of thiamine; however, this evidence only can explain the duration designed for the wash-out period. How the authors make sure that the duration of supplementation (four weeks) is long enough to be therapeutic effective in these patients. If it is possible that the negative finding resulted from the short duration of supplementation. In addition, the authors did not present the serum level of thiamine of the patients before and after thiamine supplementation. It is necessary to show the serum data of thiamine to prove the dose and duration of supplementation is effective.

2. The primary outcome of this study is a reduction of total fatigue score. As mentioned by the previous reviewers, many clinical disorders and conditions can affect the status of fatigue. The authors also have excluded patients with other comorbidities that would lead to chronic fatigue, however, many other comorbidities should also have to be excluded, such as moderate to severe degree of acute or chronic cardiac and pulmonary diseases, adrenal insufficiency, sarcopenia, obesity, malnutrition, and so on. The authors have to clarify the surveillance and exclusion criteria involved fatigue-related comorbidities.

3. Patients' nutrition status (malnutrition, obesity, sarcopenia, etc.) and daily intake amount would have big impacts on fatigue score. The authors have to present the information about daily intake of calories and protein, the nutrition status of patients before and after thiamine supplementation as well.

4. In patients with PBC, liver function has a great influence on the clinical fatigue of them. The authors also have to present the liver function of patients before and after thiamine supplementation.

5. There are some spelling mistakes that need to be corrected, such as the 3 impaired kidney fun"k"tion in figure 1.

7. PLOS authors have the option to publish the peer review history of their article (what does this mean?). If published, this will include your full peer review and any attached files.

Reviewer #1: No

Reviewer #2: No

Reviewer #3: No

---

## [Author Response · Author response to Decision Letter 1]

22 Feb 2024

Authors response to the reviewers

Reviewer #1: 

This manuscript may report as a preliminary results of clinical trial rather than a complete study. It needs more patients enrolled for consolidate findings.

Response: 

It may be true that more patients could have shown other results. However, we made a straight forward power calculation prior to study start. In addition, we would refer to our previous response to reviewer #1: 

 'We partly agree that number of patients included may be a limitation. However, the number of patients were estimated based on a specific power calculation based on our data from IBD patients. A larger study population may have revealed some effect. However, our results were not close to show a statistically significant difference. Based on the obtained results a power calculation for larger study would have demanded approximately 140 patients. Assumptions: 1) a decrease of ≥ 5 points on the PBC-40 fatigue sub-scale, 2) a within patient standard deviation of 12 points, 3) an alpha value of 0.05, and a power of 90% 4) approximately 10-15 % drop-outs. However, as we write in the discussion, the study might have been underpowered; but we also suggest that thiamine as a treatment option for PBC-fatigue could be rejected as a result of our findings.'

Reviewer #2: 

No comments from reviewer #2. 

Reviewer #3: 

Although the authors had replied the questions raised by the reviewers, there are still some points that needed to be clarified or revised.

Comment 1: 

The intervention of this study is supplementation of high dose thiamine for four weeks. The authors have explained the half life and storage time of thiamine; however, this evidence only can explain the duration designed for the wash-out period. How the authors make sure that the duration of supplementation (four weeks) is long enough to be therapeutic effective in these patients. If it is possible that the negative finding resulted from the short duration of supplementation. In addition, the authors did not present the serum level of thiamine of the patients before and after thiamine supplementation. It is necessary to show the serum data of thiamine to prove the dose and duration of supplementation is effective.

Response: 

i) Duration of treatment can be discussed. However, in other studies the effect on fatigue was obtained after 3-4 weeks. Based on these studies a thiamine exposure of 4 weeks was chosen. 

1) Bager P, Hvas CL, Rud CL, Dahlerup JF. Randomised clinical trial: high-dose oral thiamine versus placebo for chronic fatigue in patients with quiescent inflammatory bowel disease. Aliment Pharmacol Ther. 2021;53:79-86. [28 days]

2) Costantini A, Pala MI. Thiamine and fatigue in inflammatory bowel diseases: an open-label pilot study. J Altern Complement Med. 2013;19:704-8. [20 days]

ii) In this study we did not collect data on serum-thiamine. 

Firstly, because the possibility for levels below the normal range is low in a Danish population. In our IBD-study we found a few patients with levels below the normal range. However, the thiamine-effect was found not to be depended of the serum-levels. 

1) Bager P, Hvas CL, Rud CL, Dahlerup JF. Randomised clinical trial: high-dose oral thiamine versus placebo for chronic fatigue in patients with quiescent inflammatory bowel disease. Aliment Pharmacol Ther. 2021;53:79-86). 

2) Bager P, Hvas CL, Hansen MM, Ueland P, Dahlerup JF. B-vitamins, related vitamers, and metabolites in patients with quiescent inflammatory bowel disease and chronic fatigue treated with high dose oral thiamine. Mol Med. 2023;29:143.

Secondly, the dose given in this PBC-study is up to 100 fold the recommended daily dose of thiamine. Consequently, a low level of serum-thiamine at baseline will be insignificant compared to the high-dose exposure. 

Comment 2: 

The primary outcome of this study is a reduction of total fatigue score. As mentioned by the previous reviewers, many clinical disorders and conditions can affect the status of fatigue. The authors also have excluded patients with other comorbidities that would lead to chronic fatigue, however, many other comorbidities should also have to be excluded, such as moderate to severe degree of acute or chronic cardiac and pulmonary diseases, adrenal insufficiency, sarcopenia, obesity, malnutrition, and so on. The authors have to clarify the surveillance and exclusion criteria involved fatigue-related comorbidities.

Response: 

Our previous response was: 

 'We agree and since fatigue can be multifactorial, we excluded patients who had obvious other reasons for fatigue than PBC and the most frequent reasons are now mentioned in the method section. In addition, the RCT design aim to equally distribute participants who might have other reasons for fatigue.'

We have nothing more to add. We excluded patients with obvious reasons for fatigue. A complete list for possible reasons for fatigue would be endless. As stated above, the randomisation process aims to equally distribute possible differences between groups.

Comment 3: 

Patients' nutrition status (malnutrition, obesity, sarcopenia, etc.) and daily intake amount would have big impacts on fatigue score. The authors have to present the information about daily intake of calories and protein, the nutrition status of patients before and after thiamine supplementation as well.

Response: 

As stated above, we excluded patients with obvious reasons for fatigue. This would also include malnutrition. However, no one showed sign of malnutrition or severe obesity. Mean BMI was between 27-29 (Table 1). We did not monitor daily nutrition intake. And as stated above, possible differences in nutrition intake would be equally distributed via randomisation. 

Comment 4: 

In patients with PBC, liver function has a great influence on the clinical fatigue of them. The authors also have to present the liver function of patients before and after thiamine supplementation.

Response: 

Table 1 shows an equal distribution of patients between groups. This includes biochemistry. 

Comment 5: There are some spelling mistakes that need to be corrected, such as the 3 impaired kidney fun"k"tion in figure 1.

Response: 

Thank you. Good observation. Figure 1 has been updated.

---

## [Decision Letter · Decision Letter 2]

15 Mar 2024

High-dose oral thiamine versus placebo for chronic fatigue in patients with primary biliary cholangitis: A crossover randomized clinical trial

PONE-D-23-41263R2

Dear Dr. Bager,

We’re pleased to inform you that your manuscript has been judged scientifically suitable for publication and will be formally accepted for publication once it meets all outstanding technical requirements.

Kind regards,

Wan-Long Chuang, M.D., Ph.D.

Academic Editor

PLOS ONE

Additional Editor Comments (optional):

The comments are well responded. The reason for "Reject" by Reviewer 1 is the limited number of the patients. However, Reviewer 2, an expert of statistical analysis, considered the study design and patients number are acceptable. In addition, there are no other comments by the 3 reviewers. So I think this manuscript could be accepted for publication.

Reviewers' comments:

Reviewer's Responses to Questions

**Comments to the Author**

1. If the authors have adequately addressed your comments raised in a previous round of review and you feel that this manuscript is now acceptable for publication, you may indicate that here to bypass the “Comments to the Author” section, enter your conflict of interest statement in the “Confidential to Editor” section, and submit your "Accept" recommendation.

Reviewer #1: (No Response)

Reviewer #2: All comments have been addressed

Reviewer #3: All comments have been addressed

2. Is the manuscript technically sound, and do the data support the conclusions?

Reviewer #1: No

Reviewer #2: (No Response)

Reviewer #3: Yes

3. Has the statistical analysis been performed appropriately and rigorously? 

Reviewer #1: No

Reviewer #2: (No Response)

Reviewer #3: Yes

4. Have the authors made all data underlying the findings in their manuscript fully available?

Reviewer #1: Yes

Reviewer #2: (No Response)

Reviewer #3: Yes

5. Is the manuscript presented in an intelligible fashion and written in standard English?

Reviewer #1: Yes

Reviewer #2: (No Response)

Reviewer #3: Yes

6. Review Comments to the Author

Reviewer #1: The enrolled number of patients in this clinical study is not sufficient to draw solid conclusions.

Reviewer #2: (No Response)

Reviewer #3: All the comments have been answered well and the wrong spelling has been corrected.

I have no other comments to this study.

7. PLOS authors have the option to publish the peer review history of their article (what does this mean?). If published, this will include your full peer review and any attached files.

Reviewer #1: No

Reviewer #2: No

Reviewer #3: No

---

## [Editor Report · Acceptance letter]

21 Mar 2024

PONE-D-23-41263R2 

PLOS ONE

Dear Dr. Bager, 

I'm pleased to inform you that your manuscript has been deemed suitable for publication in PLOS ONE. Congratulations! Your manuscript is now being handed over to our production team.

Kind regards, 

on behalf of

Dr. Wan-Long Chuang 

Academic Editor

PLOS ONE